# Multicenter study of pneumococcal carriage in children 2 to 4 years of age in the winter seasons of 2017-2019 in Irbid and Madaba governorates of Jordan

Adnan Al-Lahham [ID]*

School of Applied Medical Sciences, German Jordanian University, Amman, Jordan

* adnan.lahham@gju.edu.jo

**Data Availability Statement:** All relevant data are within the manuscript and its supporting information files.

## Abstract

*Streptococcus pneumoniae* is one of the leading causes of death worldwide. It disseminates through colonizers and causes serious infections. Aims of this study are to determine pneumococcal carriage rate, resistance, serotype distribution, and coverage of pneumococcal conjugate vaccines from children attending day care centers from Irbid and Madaba in Jordan. Nasopharyngeal swabs were collected from day care centers (DCCs) of healthy Jordanian children 2–4 years of age from four regions of Madaba (n = 596), and from eastern Irbid (n = 423). Swabs were cultivated on Columbia blood agar base supplemented with 5% sheep blood and incubated for 18–24 hours at 37˚C with 5% $CO_2$. Alpha-hemolytic isolates were tested for optochin sensitivity and bile solubility for identification. Isolates were analyzed for antimicrobial susceptibility by the Vitek2 system and E-test (BioMérieux). Serotyping was performed using the Neufeld Quellung method. A total of 341 pneumococcal strains were isolated from 1019 nasopharyngeal (NP) samples of healthy children attending DCCs for two winter seasons from 2017–2019. Carriage rate in eastern Irbid for both seasons was 29.6% and for Madaba 37.9%. Resistance rates for Irbid and Madaba, respectively, were as follows: Penicillin (86.3%; 94.4%), erythromycin (57.0%; 78.2%), clindamycin (30.8%; 47.2%), trimethoprim-sulfamethoxazole (68.6%; 86.6%), and tetracycline (45.7%; 51.9%). Predominant serotypes for Irbid were 19F (20.8%), 23F (12.0%), 6A (10.4%), and 6B (9.6%); whereas for Madaba were 19F (24.5%), 14 (7.4%), 6A (6.9%) and 23F (6.5%). Serotype coverage of the thirteen valent pneumococcal conjugate vaccine (PCV13) was about 65% for both regions. Over 96% of isolates with PCV13 serotypes in this study were resistant to penicillin with the exception of serotypes 3 and 5. As a conclusion resistance and carriage rates among the age group 2 to 4 years reached an alarming rate especially among vaccine types, which can be controlled by pneumococcal conjugate vaccination strategies.

**Funding:** AA has received a grant from the German Jordanian University and Pfizer Pharmaceuticals Grants numbers are: Pfizer Pharmaceuticals fund number WI227419 www.pfizer.com The deanship of scientific research at the German Jordanian University under the research grant number SAMS 22/2015. www.gju.edu.jo The funders had no role in study design, data collection and analysis, decision to publish, or preparation of the manuscript.

**Competing interests:** I declare that I have no significant competing financial, professional or personal interests that might have influenced the performance or presentation of the work described in this manuscript. Furthermore, this does not alter adherence to PLOS ONE polices on sharing data and materials. No restrictions on sharing of the data/ and or materials presented in this manuscript.

## Introduction

*Streptococcus pneumoniae* is a colonizing agent of the nasopharynx, and its dissemination to other body parts can cause pneumonia, meningitis, septicemia, otitis media, and sinusitis. Main targets for pneumococcal infections are children and the elderly over 65 years of age [1–3], especially in low income countries and in countries with low diagnosis and treatment [4]. The nasopharynx is the primary reservoir for pneumococci, and its carriage is the source of disease spread between people [5]. Although colonization of *S. pneumoniae* is mostly symptomless, it is considered as a prerequisite for transmission and progress of respiratory or even systemic disease [6]. Pneumococcal carriage is believed to be an important source of horizontal spread of this pathogen within the community [7]. *S. pneumoniae* was given the name as the forgotten killer in children in 2006 by the WHO [8], which accounts for more than one third of acute bacterial sinusitis and more than one half of community-acquired bacterial pneumonia [9]. For this reason, it is essential to determine the distribution of *S. pneumoniae* serotypes among children in each country in order to address the actual value of using available commercial vaccines to minimize the pneumococcal infections [10]. Vaccination proved to be effective against pneumococcal colonization, however, a simultaneous increase in colonization with non-vaccine serotypes was observed [11, 12]. As a result, new pneumococcal conjugate vaccines (PCVs) covering 15 and 20 different serotypes are planned as the future vaccines for 2020. Predisposing factors such as children below 5 years, elderly over 65 years of age and people with low immunity can increase the pneumococcal carriage [3, 7]. Furthermore, other socio-economic factors like crowding, low income families, smoking, viral respiratory infections, lack of pneumococcal vaccination and antibiotic consumption can affect pneumococcal colonization [13]. Pneumococcal carriage rates, pneumococcal conjugate vaccine coverage and resistance rates are variable world-wide depending on the use of vaccination, correct use of antibiotics, socio-economic factors, age and urban or rural residency of people [14]. To this point, NP-carriage rate in Japanese children zero to 6 years of age attending DCCs was 43.3% with penicillin intermediate resistance of 35.7% and erythromycin resistance of 69.4% [15]. However, carriage rate in Turkish infants up to two years of age was 22.5% with 6.8% high grade penicillin resistance and 59% PCV13 coverage [16]. Another study of carriage rate in malnourished children showed 77.8% carriage in infants 6–60 months of age [17]. Studies in countries with PCVs as part of the National Immunization Programs (NIP) showed an increase of non-vaccine serotypes and decrease of vaccine serotypes. In a study performed on children in Palestine, pneumococcal carriage rate was 55.7% with dominant serotypes 6A (13.6%) followed by 19F (12.2%) [18]. Similar results were seen in Gaza on healthy children and showed pneumococcal carriage rate of 50% and 46% coverage of PCV13 [19]. Jordan is a middle-income country with about 40% of the population below 15 years of age, which is considered as large target group for the dissemination of pneumococcal infections. Pneumococcal vaccines were introduced to Jordan in the year 2000 only in the private sector, but still not part of the NIP. DCCs chosen for this study are from rural areas with low to middle income and were all not vaccinated with the PCVs. Local information about the serotypes causing diseases in young children in Jordan is essential for future vaccination programs. Continous surveillance of pneumococcal carriage and resistance is essential for preventing pneumococcal multidrug resistant strains using the PCVs [20]. The objective of this study project is to find out the prevalence of carriage with *Streptococcus pneumoniae* in the winter seasons of the years 2017–2018, and 2018–2019 in children 2–4 years of age in two district regions of Jordan (eastern Irbid and 4 regions in Madaba), and find out the resistance rates of the isolates. Monitoring serotype distribution and determination of carriage and coverage rates of different PCVs are

important criteria for appropriate application of the vaccines before introducing the PCVs into the NIP in Jordan.

## Materials and methods

### Ethics statement

The study was first approved by Independent Ethics Committee (IEC) from the Ministry of Health of Jordan followed by approval of the Ministry of Health of Jordan and approvals of each DCC director involved in this study. Informed written consent was obtained from parents of participating children, so that a parent of each child enrolled in the study gave written permission for a nasopharyngeal swab to be taken from their child and for relevant data to be used for protocoling. The parents were also educated on the benefits of the future vaccination with the available PCVs. Questionnaires with names, date of birth, gender, number of household, address, and history of PCV vaccination were recorded at the time of sample collection. All NP-samples were collected from trained medical doctors of each DCC. Positive results of carriage with resistance analysis and serotyping were sent to the medical doctors of each DCC to be registered on files of the participating children.

### Study period and population involved

Surveillance study of nasopharyngeal carriage and antibiotic resistance of *Streptococcus pneumoniae* in healthy Jordanian infants was launched in two main cities during two winter seasons in the period between October 2017 to the end of March 2018 and from October 2018 to the end of March 2019. The study was conducted on 1019 healthy children 2–4 years of age. Samples were collected from four DCCs of Madaba (East, west, north and south), and one DCC of eastern Irbid. All DCCs are found in rural areas with low to middle income families and the participating children have no history of PCV vaccination. The municipality of Madaba has a population of 82,335 inhabitants and for Irbid 307,480 inhabitants. Age group 2 to 4 years constitute of 6.6% of the total population of Jordan. All children coming to the day care centers (DCCs) for check-ups and for routine consultations in the age group 2–4 years were enrolled in this study. During the whole study period, a single nasopharyngeal swab was taken from each child. Duplicate swabs from the same child in different periods of the study were not considered.

### Sampling and identification

Nasopharyngeal cotton swabs were collected as previously described [21]. Swabs were placed in Stuart transport media and were put in transport box for direct transport to the microbiology reference lab of the German Jordanian University. Samples were directly streaked on Columbia blood agar base supplemented with 5% sheep blood and incubated for 24 hours at 37˚C with 5% $CO_2$. Suspected α-haemolytic colonies were further tested for optochin sensitivity (bioMérieux) and bile solubility [22–24]. Positive isolates sensitive to optochin and bile soluble were frozen at -70˚C for serotyping and antimicrobial sensitivity testing.

### Capsular serotyping

Pure cultures of *Streptococcus pneumoniae* incubated for 18–24 hours at 37˚C with 5% $CO_2$ were taken for serotyping by the Neufeld's Quellung reaction method using commercially available type and factor sera provided by the Statens Serum Institute, Copenhagen, Denmark [25].

## Antimicrobial susceptibility testing

Minimal inhibitory concentration (MIC) testing was performed using the micro broth dilution method with the VITEK2 compact system using cards AST03 for *Streptococcus pneumoniae*, and using E-tests obtained from Oxoid and bioMérieux. Antibiotics used were: Penicillin G (PEN), amoxicillin (AMOX), cefotaxime (CETA), cefuroxime (CEFU), cefpodoxim (CEPO), ceftriaxone, clarithromycin (CLA), clindamycin (CLI), tetracycline (TET), levofloxacin (LEVO), moxifloxacin, telithromycin, sulfamethoxazole-trimethoprim (SXT), chloramphenicole (CHA), vancomycin, tigecycline, linezolid, and rifampicin. *S. pneumoniae* ATCC 49619 was used as a control strain. Breakpoints for susceptibility used in this study were interpreted according to Clinical Laboratory Standards Institute (CLSI) guidelines [26].

## Primary and secondary efficacy endpoints

This include the frequency of NP-carriage, serotype distribution and antimicrobial resistance patterns of the strains in children between 2–4 years of age. Although there is no true efficacy in this project, determination and assessment of the vaccine type pneumococcal carriage was determined. Furthermore, rates of resistant strains were documented.

## Statistical analysis

Student t-test was considered for significant differences using 2-tailed values with the level of significance at $p < 0.05$. Other analysis include: Rate of carriage, vaccine and non-vaccine serotype coverage, resistance rates to antibiotics, Analysis for age, sex, and seasonal variations.

## Results

The total number of NP-samples collected were 1019, where 530 (52.0%) were from male and 489 (48.0%) were from females. Male swabs were 40.2% from Irbid and 59.8% from Madaba DCCs, whereas female swabs were 42.9% from Irbid and 57.1% from Madaba DCCs (Table 1).

Detailed number of NP-samples from each DCC and each season is presented in S1 Table.

All children enrolled in the study were in the age group 2 to 4 years and the average age of all children at the time of enrolment was 36.2 months. Total carriage rate in all centers for both cities and both seasons was 33.5%. A significant difference ($P < 0.05$) was noticed for carriage in both seasons between Irbid (29.6%), and all DCCs of Madaba (36.2%). Gender significant difference ($P < 0.05$) in the study was also noticed between male carriers of all centers of Madaba (41.6%) compared to male carriers (23.8%) from Irbid. Total carriage for females for all centers in Madaba was the same in both seasons (33.0%). Total carriage rate in Irbid in W2017-2018 (29.0%) was not significantly different from the carriage rate in the W2018-2019 (29.8%) ($P < 0.05$). Detailed numbers of gender carriage for each center and each season are available in the supplementary data, (see S2 Table).

The average household measured in this study for eastern Irbid was 5.2 children for each family, whereas in Madaba was 3.6 children. Carriage rates for Irbid and Madaba for

**Table 1. Source of 1019 nasopharyngeal samples according to the gender in the winter seasons of 2017–2018 and 2018–2019.**

| City | Male n (%) | Female n (%) | Total n (%) |
|---|---|---|---|
| Easten Irbid | 213 (40.2%) | 210 (42.9%) | 423 (41.5%) |
| Madaba all regions | 317 (59.8%) | 279 (57.1%) | 596 (58.5%) |
| Total | 530 (52.0%) | 489 (48.0%) | 1019 (100%) |

**Table 2. Carriage in Irbid and Madaba with household of > = 5 or less than 5 children for both winter seasons.**

| Household | Carriage in Irbid | | Carriage in Madaba | |
|---|---|---|---|---|
| | n | % | n | % |
| > = 5 children | 83/125 | 66.4 | 152/216 | 70.4 |
| < 5 children | 42/125 | 33.6 | 64/216 | 29.6 |

households with five or more children was shown to be 66.4% for Irbid, and 70.4% for Madaba. (Table 2).

In both cities, carriage was significantly higher (*P<0.05*) in the age group 2–3 years old children, which was 32.6% for Irbid and 38.1% for Madaba. Penicillin resistance in this age group has reached more than 96% in both cities, and that the coverage of PCV13 was 68.3% for Irbid and 65.5% for Madaba. Detailed numbers are available in supplementary file (see S3 Table).

On a monthly basis, pneumococcal carriage was highest among strains isolated in March (44.7%) for eastern Irbid, which was not significantly different (*P > 0.05*) from those isolated from Madaba (45.5%) as shown in Figs 1 and 2. Detailed numbers for Figs 1 and 2 are available in supplementary file (see S4 and S5 Tables). Coverage of PCV13, on the other hand, has reached 100% for strains isolated in November followed by 78.6% for strains isolated in January for eastern Irbid (Fig 1). The same highest coverage rate was noticed for the strains isolated in November for Madaba (78.9%) (Fig 1). With regard to the resistance rate on monthly basis for both regions, penicillin resistance was highest in the months from October to January (96.8%-100%) for eastern Irbid, but was highest in February (100%) for Madaba as shown in Fig 2. Furthermore, clarithromycin resistance reached 100% in November for Irbid followed by 90.9% in February, and that trimethoprim-sulfamethoxazole resistance was highest in isolates of November and December of the year with rates 96.8%-100%. As a comparison

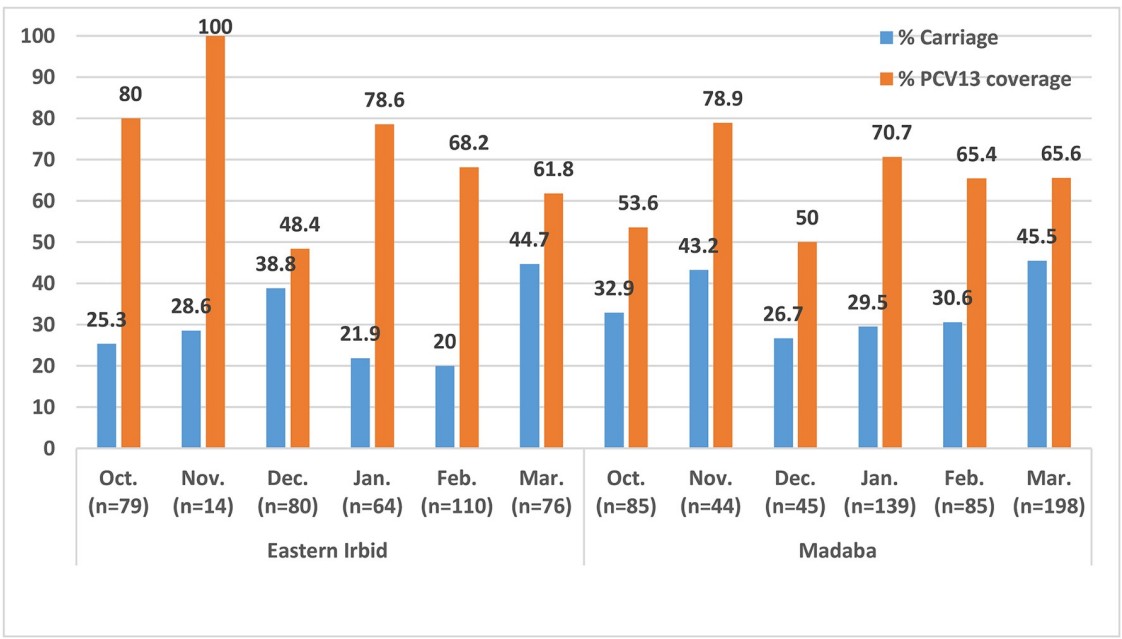

**Fig 1. Distribution of carriage and PCV13 coverage rates according to the month of isolation from Irbid and Madaba in both winter seasons.**

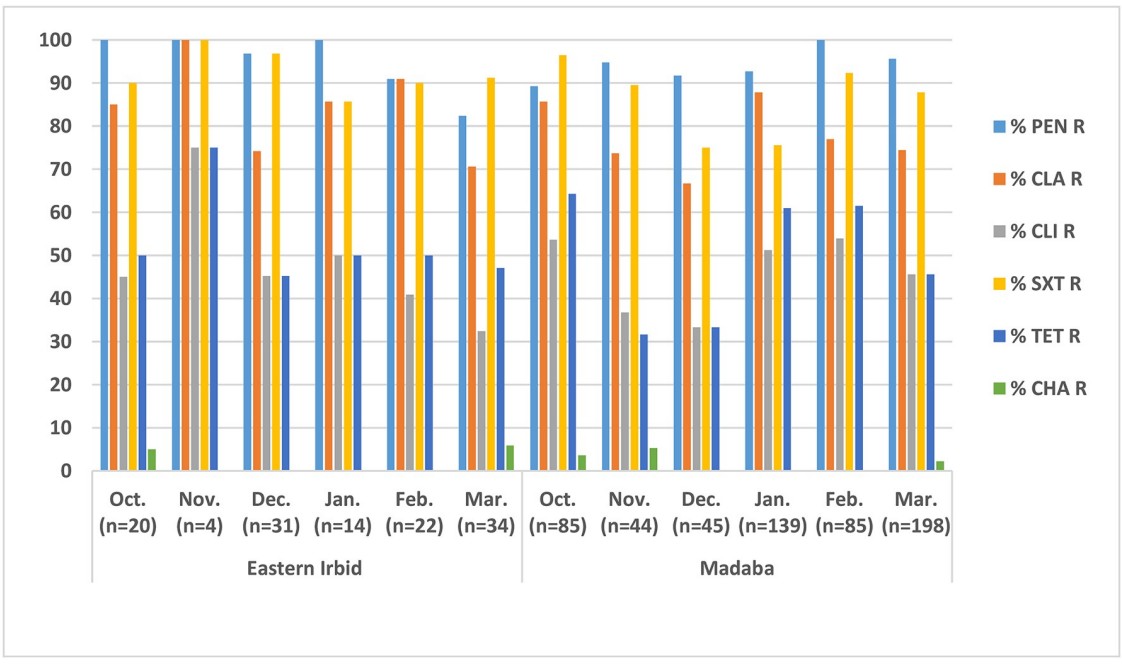

**Fig 2. Resistance rates of pneumococcal isolates on monthly basis from eastern Irbid and Madaba in both winter seasons.**
Abbreviations: PEN (Penicillin), CLA (Clarithromycin), CLI (Clindamycin), SXT (Sulfamethoxazole-Trimethoprim), TET (Tetracycline), CHA (Chloramphenicol).

clarithromycin resistance was highest in isolates of January (87.8%), but trimethoprim-sulfamethoxazole resistance was highest in isolates of October (96.4%) for Madaba (Fig 2).

The overall rate of carriage for Madaba regions (36.2%) was significantly higher ($P < 0.05$) than eastern Irbid (29.6%), however vaccine types of isolates from eastern Irbid were 66.4%, compared to 67.1% for all regions of Madaba, although southern region of Madaba showed the highest coverage of 73.2% (Table 3). Differences in carriage of *Streptococcus pneumoniae* isolates from each DCC of Madaba regions showed highest carriage of 38.4% in eastern Madaba and lowest of 31.6% in northern Madaba, with an average carriage of all Madaba regions of 36.2% for both seasons as shown in Table 3.

Resistance rates for penicillin, cefotaxime, clarithromycin, trimethoprim-sulfamethoxazole and tetracycline for Irbid were 92.8%, 0.8%, 79.2%, 91.2% and 48.8%, respectively. As a comparison, in all regions of Madaba resistance rates were 94.5%, 5.1%, 78.2%, 86.5% and 51.9%,

**Table 3. Rate of carriage of *Streptococcus pneumoniae* from Irbid and Madaba DCCs for both winter seasons of W2017-18 and W2018-19 with vaccine types and non-vaccine types.**

| City | DCC | Carriage for both seasons | | Vaccine types of PCV13 | | Non-Vaccine types | |
|---|---|---|---|---|---|---|---|
| | | n | % | n | % | n | % |
| Irbid | Alrazi | 125/423 | 29.6 | 83/125 | 66.4 | 42/125 | 33.6 |
| Madaba | East | 96/250 | 38.4 | 66/96 | 68.8 | 30/96 | 31.3 |
| | West | 34/94 | 36.2 | 22/34 | 64.7 | 12/34 | 35.3 |
| | North | 30/95 | 31.6 | 16/30 | 53.3 | 14/30 | 46.7 |
| | South | 56/157 | 35.7 | 41/56 | 73.2 | 15/56 | 26.8 |
| All DCCs of Madaba | | 216/596 | 36.2 | 145/216 | 67.1 | 71/216 | 32.9 |
| Total | | 341/1019 | 33.5 | 228/341 | 66.9 | 113/341 | 33.1 |

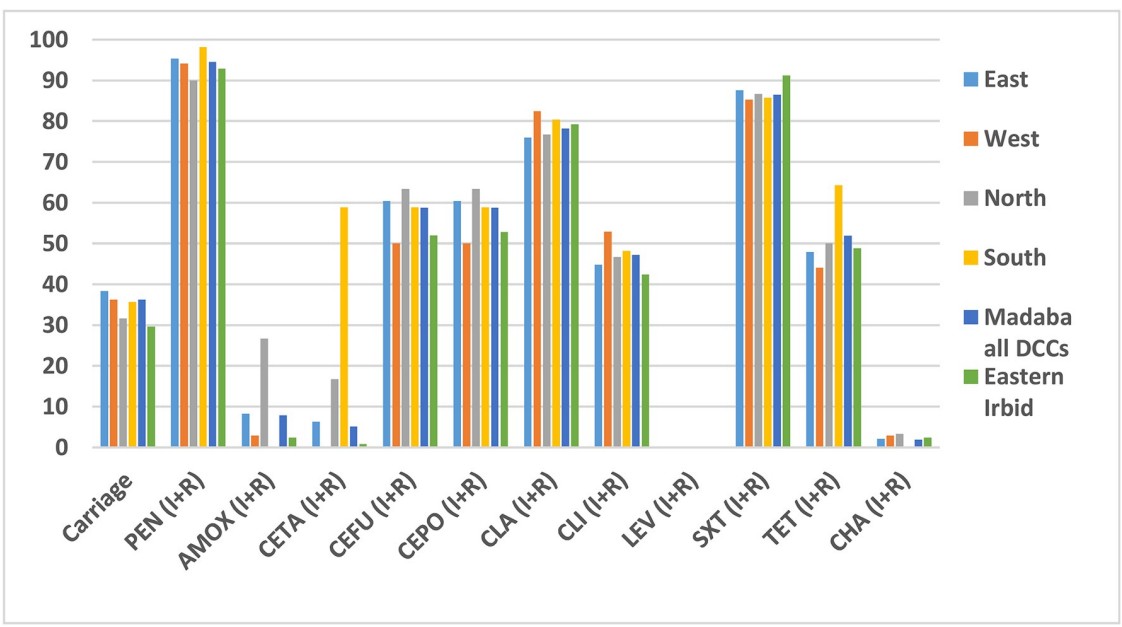

**Fig 3. Carriage and resistance rates of *Streptococcus pneumoniae* isolates from eastern Irbid and all DCCs of Madaba for both winter seasons from October 2017 to End of March 2019.** Abbreviations: PEN: Penicillin; AMOX: Amoxicillin; CETA: Cefotaxime; CEFU: Cefuroxime; CEPO: Cefpodoxime; CLA: Clarithromycin; CLI: Clindamycin; LEV: Levofloxacin; SXT: Trimethoprim-sulfamethoxazole; TET: Tetracycline; CHA: Chloramphenicole; I: Intermediate resistance; and R: Resistant.

respectively (Fig 3). Highest resistance rates for penicillin and tetracycline in both winter seasons was detected in the south of Madaba with rates 98.2% and 64.3%, respectively. For both seasons, trimethoprim-sulfamethoxazole resistance was highest in Irbid followed by eastern Madaba with resistance rates 91.2% and 87.6, respectively. Detailed numbers for Fig 3 is available in supplementary file (see S6 Table). No resistance was noticed for the following antibiotics: Moxifloxacin, levofloxacin, telithromycin, vancomycin, ceftriaxone, tigecycline, rifampicin and linezolid.

Coverage of the pneumococcal conjugate vaccines PCV7, PCV10, and PCV13 for the 5 DCCs of the study showed highest coverage for the PCV13 in South of Madaba with coverage of 73.2% followed by eastern Irbid DCC with PCV13 coverage of 65.6% (Fig 4). However, coverage of available pneumococcal vaccines PCV7, PCV10 and PCV13 in Madaba regions showed highest coverage for both seasons in Madaba west with PCV7 and PCV10 coverage of 52.9% and 55.9%, respectively. Furtheremore, in Madaba regions coverage rates for PCV7, PCV10 and PCV13 in both winter seasons were 49.5%, 50.0%, and 65.3%, respectively. These coverage rates for both cities in both winter seasons were not significantly different (*P> 0.05*).

Predominant serotypes for both winter seasons for eastern Irbid city were: 19F, 23F, 6A, and 14 with rates 20.8%, 12%, 10.4%, 9.6%, and 7.2%, respectively. However, predominant serotypes for all Madaba regions in both winter seasons were 19F, 14, 6A, 23F and 6B with rates 24.5%, 7.4%, 6.9%, 6.5%, and 6.0%, respectively (Table 4). In both cities, the predominant serotypes were all covered by the PCVs and were the same with different rates and rankings.

Antibiotic resistance of vaccine serotypes was highest for penicillin with exception of the serotypes 3 and 5 (Table 5). Furthermore antibiotic resistance of non-vaccine serotypes showed less resistance to the majority of antibiotics compared to vaccine types (Table 5).

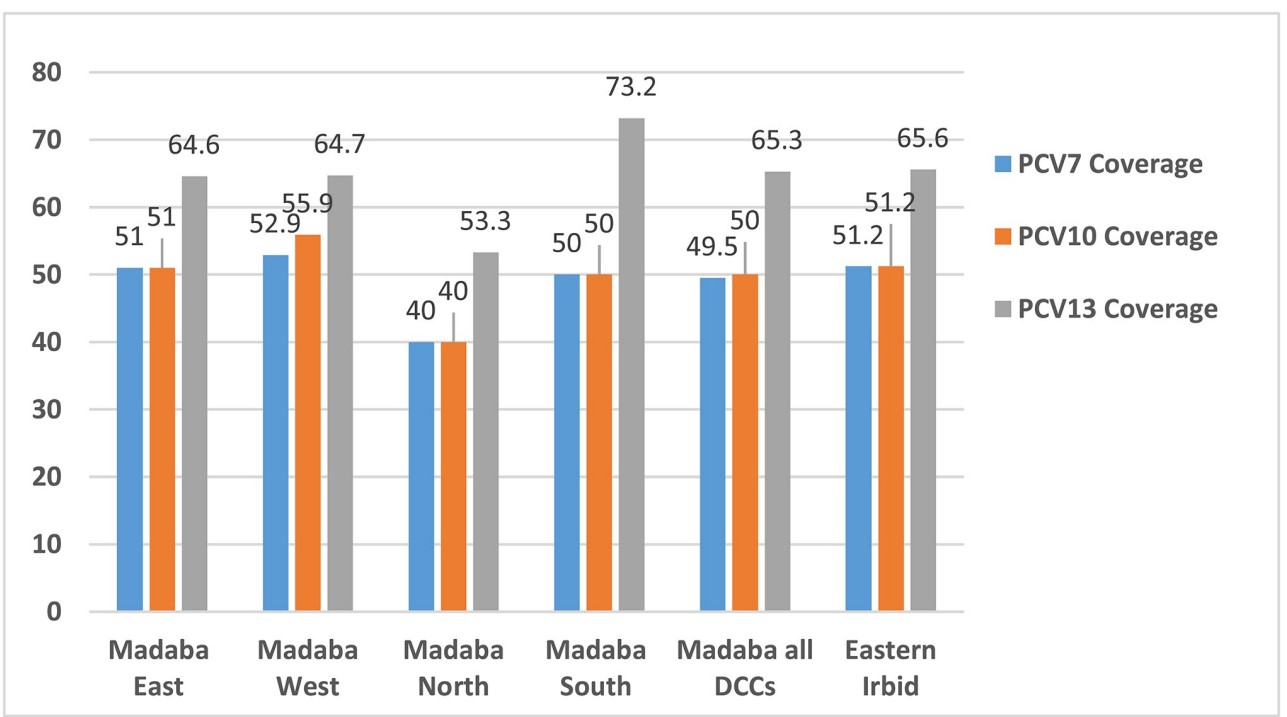

**Fig 4. Coverage of the pneumococcal conjugate vaccines in each DCC for both winter seasons from October 2017-end of March 2019.**

## Discussion

A total of 1019 children were enrolled in the study from 5 day care centers in two main cities of Jordan, namely eastern Irbid and Madaba. All of the children enrolled in this study had no history of previous vaccination with the pneumococcal conjugate vaccines, since families with low to middle income and cannot afford the cost of PCV at the private sector and that the PCV is not available in the National Immunization Program of the country. This is the largest number of the age group 2–4 years taken for studying the carriage of *S. pneumoniae* in two winter seasons of the years 2017–2018 and 2018–2019, which provides an insight into the state of pneumococcal carriage among this age group of healthy children attending day-care centers. Our results support the well established findings that acquisition of new strains of *S. pneumoniae* appears to be a seasonal phenomenon, with highest rates in the winter months [27, 28]. Furthermore, pneumococcal carriage proved to be more common among children 2 years of age with carriage rate of 70% as found in Norway [29]. There are variations of carriage between male and female genders among the DCCs for both seasons, in all regions of Madaba, males show a higher carriage rate than females with the highest rate in east Madaba, where 43.2% of males were carriers. Males at all Madaba DCCs prove to have higher carriage rates than females. This result was the opposite in Irbid, where female carriage rate was 31.0% compared to male carriage rate 28.2%. Few studies have been done on sex disparities in pneumococcal carriage or in pneumococcal invasive diseases, these disparities are thought to be due to biological and behavioral differences, which also vary with age [30]. Factors that contribute to high carriage rates in both cities are: low income, family members that smoke, and a high number of children per household as an important risk factor [4, 31]. Families with household more than 5 children showed carriage rate of 70.4% for Madaba, and 66.4% for eastern Irbid. The overall pneumococcal carriage prevalence in this study was 33.5% for all centers in both

**Table 4. Percentage of serotypes recovered from all DCCs for both winter seasons.**

| Serotype | East (n = 96) | West (n = 34) | North (n = 30) | South (n = 56) | Madaba all DCCs (n = 216) | Eastern Irbid (n = 125) |
|---|---|---|---|---|---|---|
| 3 | 3.1 | 2.9 | 0 | 5.4 | 3.2 | 0.8 |
| 5 | 0 | 2.9 | 0 | 0 | 0.5 | 0 |
| 6A | 4.2 | 2.9 | 13.3 | 10.7 | 6.9 | 10.4 |
| 6B | 7.3 | 0 | 10 | 5.4 | 6 | 9.6 |
| 6C | 1.0 | 0 | 0 | 0 | 0.5 | 1.6 |
| 7B | 0 | 0 | 0 | 1.8 | 0.5 | 0 |
| 9N | 0 | 0 | 3.3 | 0 | 0.5 | 0 |
| 9V | 0 | 0 | 0 | 5.4 | 1.4 | 0.8 |
| 10A | 1.0 | 0 | 3.3 | 1.8 | 1.4 | 0.8 |
| 11A | 3.1 | 8.8 | 6.7 | 0 | 4.2 | 3.2 |
| 14 | 6.3 | 11.8 | 10 | 5.4 | 7.4 | 7.2 |
| 15A | 0 | 2.9 | 0 | 0 | 0.5 | 2.4 |
| 15B | 1.0 | 0 | 0 | 5.4 | 1.9 | 2.4 |
| 15C | 1.0 | 8.8 | 0 | 3.6 | 2.8 | 4 |
| 16A | 2.1 | 0 | 0 | 0 | 0.9 | 0 |
| 16F | 1.0 | 0 | 0 | 0 | 0.5 | 3.2 |
| 17F | 1.0 | 2.9 | 0 | 1.8 | 1.4 | 0.8 |
| 18C | 3.1 | 0 | 0 | 7.1 | 3.2 | 0.8 |
| 19A | 6.3 | 2.9 | 0 | 7.1 | 5.1 | 3.2 |
| 19F | 28.1 | 26.5 | 13.3 | 23.2 | 24.5 | 20.8 |
| 23A | 3.1 | 5.9 | 6.7 | 3.6 | 4.2 | 1.6 |
| 23F | 5.2 | 14.7 | 6.7 | 3.6 | 6.5 | 12 |
| #23 | 4.2 | 0 | 0 | 0 | 1.9 | 0.8 |
| 28A | 3.1 | 0 | 6.7 | 1.8 | 2.8 | 0.8 |
| 35A | 1.0 | 0 | 0 | 0 | 0.5 | 0.8 |
| NT | 5.2 | 0 | 3.3 | 3.6 | 3.7 | 1.6 |
| Mixed 14 & 6B | 1.0 | 0 | 0 | 0 | 0.5 | 0 |
| Others | 7.3 | 5.9 | 16.7 | 1.8 | 6.9 | 10.8 |

*Others: Serotypes belong to the group sera C, D, E, F, G, and not involved in the PCVs.

#: Serogroup 23 was negative for all subtypes b, c and d.

seasons. This is consistent with previous carriage study of healthy children not exposed to PCVs of the same age group conducted in Wadi Al Seer of Jordan (33.3%) [2]. For eastern Irbid, carriage rate for both winter seasons was 29.6%, which is significantly less than carriage rate of all Madaba DCCs in both seasons (36.2%) (P< 0.05). Similar studies were performed in Jordan in other cities, and showed 55.1% for Wadi Al Seer [2], and 58.1% for Ajlun [5]. In both Wadi Al Seer and Ajlun, at least two to three samples were taken from each case, therefore a higher carriage rate was seen. This is also consistent with the study performed by Gray *et al.* 1980 [27], where more than one sample improved the isolation rate of *S. pneumoniae*. Overall in both cities, 65.4% of pneumococci belonged to PCV13 serotypes.

Due to limited serotyping data available from invasive pneumococcal disease in Jordan, monitoring serotype changes in pneumococcal carriage may be a practical way of assessing vaccine impact. The distribution of pneumococcal serotypes varied among regions of Madaba and eastern Irbid, ranging from 65.6% of serotypes carried by children in eastern Irbid belonging to PCV13 serotypes compared to 65.3% for children in Madaba. The proportion of circulating pneumococci with serotypes covered by the vaccine may vary among regions. The

**Table 5. Antibiotic resistance for the vaccine serotypes and non-vaccine serotypes in both cities from October 2017 to the end of March 2019.**

| Serotype | % PEN R | % CLA R | % CLI R | % LEV R | % SXT R | % TET R | % CHA R |
|---|---|---|---|---|---|---|---|
| 3 (n = 8) | 37.5 | 12.5 | 0 | 0 | 37.5 | 12.5 | 0 |
| 5 (n = 1) | 0 | 0 | 0 | 0 | 100 | 0 | 0 |
| 6A (n = 28) | 96.4 | 92.9 | 32.1 | 0 | 78.6 | 32.1 | 0 |
| 6B (n = 25) | 100 | 96 | 72 | 0 | 96 | 76 | 8 |
| 9V (n = 4) | 100 | 100 | 0 | 0 | 100 | 100 | 0 |
| 14 (n = 26) | 100 | 100 | 92.3 | 0 | 92.3 | 34.6 | 0 |
| 18C (n = 8) | 100 | 0 | 0 | 0 | 87.5 | 2.4 | 0 |
| 19A (n = 15) | 100 | 93.3 | 6.7 | 0 | 93.3 | 13.3 | 0 |
| 19F (n = 79) | 100 | 98.7 | 91.1 | 1.3 | 91.1 | 88.6 | 0 |
| 23F (n = 29) | 100 | 82.8 | 34.5 | 3.4 | 100 | 48.3 | 13.8 |
| 6C (n = 3) | 66 | 100 | 66 | 0 | 66 | 100 | 0 |
| 7B (n = 1) | 0 | 100 | 100 | 0 | 0 | 100 | 0 |
| 9N (n = 1) | 100 | 100 | 100 | 0 | 100 | 100 | 0 |
| 10A (n = 4) | 100 | 0 | 0 | 0 | 100 | 25 | 0 |
| 11A (n = 13) | 100 | 69.2 | 7.7 | 0 | 100 | 7.7 | 0 |
| 15A (n = 4) | 100 | 100 | 100 | 0 | 100 | 100 | 0 |
| 15B (n = 7) | 100 | 100 | 14.3 | 0 | 100 | 85.7 | 0 |
| 15C (n = 11) | 100 | 90.9 | 0 | 0 | 100 | 90.9 | 0 |
| 16A (n = 2) | 50 | 0 | 0 | 0 | 50 | 0 | 0 |
| 16F (n = 5) | 80 | 60 | 20 | 0 | 100 | 40 | 0 |
| 17F (n = 4) | 100 | 25 | 0 | 25 | 75 | 0 | 0 |
| 23A (n = 11) | 100 | 72.7 | 36.4 | 0 | 72.7 | 27.3 | 0 |
| #23 (n = 5) | 100 | 20 | 0 | 20 | 60 | 0 | 0 |
| 28A (n = 7) | 85.7 | 85.7 | 42.9 | 0 | 85.7 | 42.9 | 0 |
| 35A (n = 2) | 100 | 50 | 50 | 0 | 100 | 0 | 0 |
| NT (n = 10) | 80 | 80 | 10 | 10 | 90 | 50 | 0 |
| *others (n = 28) | 71.4 | 28.6 | 10.7 | 3.6 | 78.6 | 14.3 | 3.6 |

Abbreviations: Pen (Penicillin), CLA (Clarithromycin), CLI (Clindamycin), LEV (Levofloxacin), SXT (Trimethoprim-sulfamethoxazole), TET (Tetracycline), CHA (Chloramphenicole), R (Intermediate and high grade resistance), NT (non-typeable).

#: serogroup 23 (was positive with the serogroup 23, but negative with type sera b, c and d).

* Others (serotypes belong to the old serogroups C, D, E, F, G, and are non-vaccine types).

methodological differences in carriage studies and variations of carriage estimates may result from differences in climate, season or crowding which influence transmission [32]. It was found that children less than 5 years of age have more carriage of pneumococci than children 6–13 years of age [33].

High resistance rates were observed in this study, reaching 93.8% for penicillin, 78.6% for clarithromycin, 45.5% for clindamycin, 88.3% for co-trimoxazole, and 50.7% for tetracycline, for all DCCs in both cities and both seasons. These resistance rates were variable among DCCs and in each season. High consumption of antibiotics in the country, and a history of antibiotic consumption prior to their visits to the DCC could be the reason or contribute to increased resistant strains [34, 35]. Serotypes prevailing in Irbid were 19F (20.8%), 23F (12.0%), 6A (10.4%), and 6B (9.6%); whereas for Madaba were 19F (24.5%), 14 (7.4%), 6A (6.9%) and 23F (6.5%). In a study performed in China in 2008 for hospitalized pediatric patients younger than 14 years of age 94.7% of isolates were multidrug resistant, and 86.1% were of the PCV13

vaccine types, and that dominant serotypes were 19F (31.6%), 19A (19.8%), 23F (11.2%), 6A (9.1%), 14 (9.1%) and 15B (5.9%) [36].

## Conclusions

Continued surveillance of pneumococcal carriage, resistance and serotype distribution is essential, especially in countries where PCV is not included in the National Immunization program. In this study pneumococcal carriage rate and resistance rates were high taking into consideration that only one NP-swab was taken from each child. This was also true in households with more than 5 children. Resistance to antibiotics was highest for penicillin and trimethoprim-sulfamethoxazole and that the coverage rates of PCVs were high in some regions. Vaccine-serotype pneumococcal colonization continue to be common in children and proved to possess more resistance than non-vaccine serotypes. National vaccination using PCVs would likely prevent episodes of *S. pneumoniae* diseases and carriage together inhibiting the spread of antibiotic resistance. These data should help public health authorities in making future strategies to prevent pneumococcal carriage and resistance by choosing the appropriate PCV to cover most of serotypes rotating in the community.

## Supporting information

**S1 Table.**
(XLSX)

**S2 Table.**
(XLSX)

**S3 Table.**
(XLSX)

**S4 Table.**
(XLSX)

**S5 Table.**
(XLSX)

**S6 Table.**
(XLSX)

## Acknowledgments

The author would like to thank the Ministry of Health, ethical committee, IRB, and all health directories and centers that facilitated this study. The author would also like to thank the families of the children involved in this study and the author would like to extend special thanks to the research assistant Noor Bataineh for help with the VITEK2 compact and Dr Ibrahim Alkhdour from Madaba, and staff nurse Arwa Obeidat from Irbid for swab collection. The author acknowledges Pfizer Pharmaceuticals and the deanship of scientific research at the German Jordanian University for their support.

## Author Contributions

**Conceptualization:** Adnan Al-Lahham.

**Data curation:** Adnan Al-Lahham.

**Formal analysis:** Adnan Al-Lahham.

**Funding acquisition:** Adnan Al-Lahham.

**Investigation:** Adnan Al-Lahham.

**Methodology:** Adnan Al-Lahham.

**Project administration:** Adnan Al-Lahham.

**Resources:** Adnan Al-Lahham.

**Software:** Adnan Al-Lahham.

**Supervision:** Adnan Al-Lahham.

**Validation:** Adnan Al-Lahham.

**Visualization:** Adnan Al-Lahham.

**Writing – original draft:** Adnan Al-Lahham.

**Writing – review & editing:** Adnan Al-Lahham.

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
