## [Decision Letter · Decision Letter 0]

3 Apr 2020

PONE-D-20-05366

Multicenter study of pneumococcal carriage in children 2 to 4 years of age in the winter seasons of 2017-2019 in Irbid and Madaba governorates of Jordan

PLOS ONE

Dear Dr. Al-Lahham,

Thank you for submitting your manuscript to PLOS ONE. After careful consideration, we feel that it has merit but does not fully meet PLOS ONE’s publication criteria as it currently stands. Therefore, we invite you to submit a revised version of the manuscript that addresses the points raised during the review process.

Both the reviewers had problem with the presentation of data and suggested to use graphs/charts to depict major findings of the study. I agree with the reviewers and suggest that authors summaries the data in graphs and detailed tables be provided as supplementary material.

We would appreciate receiving your revised manuscript by May 18 2020 11:59PM. To enhance the reproducibility of your results, we recommend that if applicable you deposit your laboratory protocols in protocols.io, where a protocol can be assigned its own identifier (DOI) such that it can be cited independently in the future. For instructions see: http://journals.plos.org/plosone/s/submission-guidelines#loc-laboratory-protocols

We look forward to receiving your revised manuscript.

Kind regards,

Anirudh K. Singh, Ph.D

Academic Editor

PLOS ONE

Journal Requirements:

"AA has received a grant from the German Jordanian University and Pfizer Pharmaceuticals

Grants numbers are:

Pfizer Pharmaceuticals fund number WI227419

www.pfizer.com

The deanship of scientific research at the German Jordanian University under the research grant number SAMS 22/2015.

www.gju.edu.jo

The funders had no role in study design, data collection and analysis, decision to publish, or preparation of the manuscript"

We note that you received funding from a commercial source: 'Pfizer Pharmaceuticals'

4. Please include your tables as part of your main manuscript and remove the individual files. Please note that supplementary tables (should remain/ be uploaded) as separate "supporting information" files

Reviewers' comments:

Reviewer's Responses to Questions

**Comments to the Author**

1. Is the manuscript technically sound, and do the data support the conclusions?

Reviewer #1: Yes

Reviewer #2: Partly

2. Has the statistical analysis been performed appropriately and rigorously? 

Reviewer #1: Yes

Reviewer #2: No

3. Have the authors made all data underlying the findings in their manuscript fully available?

Reviewer #1: Yes

Reviewer #2: Yes

4. Is the manuscript presented in an intelligible fashion and written in standard English?

Reviewer #1: Yes

Reviewer #2: No

5. Review Comments to the Author

Reviewer #1: Summary:

This study describes the distribution of carriage rates of S. pneumoniae over a two year period in children under 5 years of age attending day care centres in two cities of Jordan. The information presented add to current knowledge regarding the global distribution of carriage S. pneumoniae. The study also presents antimicrobial resistance information related to these carriage isolates.

Major Issues:

A major concern is the confusing nature of how the results presented. Many of the finer details can be summarized in the text referring to charts for support, highlight interesting differences for the reader to note. The numerous tables presenting the finer details can be presented in supplementary materials.

Minor Issues:

Page 2, line 35: “DDC” should be defined on line 27.

Page 2, line 41: “PCV13 “ should be defined.

Page 2, line 41: “PCV13 serotype coverage for Irbid was 65.6%, and for all regions of Madaba 65.3%.” Since the rates are pretty much the same, for the abstract I suggest something similar to “Serotype coverage of the 13 valent pneumococcal vaccine (PCV13) was about 65% for both regions.”

Page 2, line 42: “All vaccine serotypes isolated in this study included in the PCV13 showed penicillin resistance from 96.4% to 100% with exception to serotypes 3 and 5 in both cities.” Suggest re-wording to: “Over 96% of isolates with PCV13 serotypes in this study were resistant to penicillin with the exception of serotypes 3 and 5.”

Page 3, line 49: “… (S. pneumoniae)…” is not required here.

Page 6, line 118: Describe what the level of significance was, ie p<0.05 using 2-tailed values.

Page 7, lines 135-138: If the differences between the two cities are not analyzed, the manuscript can be shortened by combining these data for both cities. Keep these details in table 1. Perhaps make table 1 a supplementary table?

Page 8 onward; page number missing from draft manuscript.

Line 151: Suggest using a chart to display this information and put Table 2 into supplementary material.

Line 165: Table 4 not included in draft. Again, maybe use a chart for this information instead of a table?

Line 171: If there is no statistical difference, just state as such and combine data for both cities. State detailed data only for the significant differences. The P<0.05 indicates significance? This should be p>0.05?

Line 172: I would suggest separating the monthly coverage variations of coverage from that of the antimicrobial resistance data. Keep the same data type together (ie. 1)monthly/seasonal trend of coverage; 2) AMR distributions geographically/temporal). Furthermore, perhaps the monthly trend of AMR is not needed, but present the just annual /regional differences if present?

Lines 200-203: Data in Tables 8-10 can be summarized to show the AMR results into a chart instead of the data from tables 6-7? Put detailed counts into supplementary material.

Line 221-223: Again, for easier reading, if there is no large or interesting differences, just make a general statement with the details.

Lines 232-240: Tables 8-10 seem to repeating data covered in previous tables? It is getting confusing as to what differences the read should be noting. Are tables 10-11 necessary? Can the data summarized and combined with previous data tables in a more efficient manner?

Lines 254- 278: Tables 11 – 15 should be summarized into a figure if possible.

Line 283: Spelling error: “… studding…” ; “studying” ?

Reviewer #2: Summary:

In this study the author has given statistics on the pneumococcal carriage rate in children of 2 to 4 years of age during winter season of 2017-2019 in Irbid and Madaba governorates of Jordan. Alongside, information regarding resistance, serotype distribution, and coverage of pneumococcal conjugate vaccines for the same study population is also provided.

Recommendations:

Accept after major revision. I support the potential publication of this manuscript due to its scientific interest. On the other hand, many aspects of the manuscript need to be extensively improved. I suggest the author to consider all of the following major remarks to improve the quality of the presentation of his work. To reiterate, the quality of the manuscript must be strengthened for it to be finally accepted.

1) The use of English must greatly improve if the paper is to be published in PLOS ONE. The author needs to take inputs from similar papers in order to improve the overall presentation of his work.

2) The “Introduction” section of the manuscript requires extensive revision. First of all, the author needs to expand the review of literature that is relevant to his study. Second, the aim of the study needs to be properly highlighted and justified. Instead of setting aim in the frame of a simplistic question (reviewer’s personal point of view), I would suggest that the author should attempt to present the key objectives of his study with regards to what is currently known (i.e. literature), thus highlighting the added value of the paper.

3) The inclusion criteria for the children must be clearly stated (for e.g., age group, previous vaccination history considerations and equal population from both the genders).

4) Details of the demographic surveillance site with independent survey information (Socio-demographic characteristics) should be clearly described.

5) There is no mention of the sampling procedures followed or details of the sample size calculation for this study.

6) To my point of view, materials and method section of the manuscript is written in an amateurish way and does not match the quality standards for being publish in PLOS ONE. The description of the data collection, transportation, laboratory analysis, variable definition and data management & analysis is almost chaotic, while the use of terminology and language is too simplified. Probably a detail flowchart of methodology including the experiment results should be mentioned. The protocol for sampling from 1019 to 341 strain should be elaborated.

7) To my view, this is not a proper presentation of the statistical results for a survey study. Again, information provided is very hard to follow. The author should consult similar papers to view how the data of a huge sampling study over 2 years should be properly presented and justified. For example, the data given in the manuscript are mainly in tabular form but can be more informative, if represented in the form of graphs or charts. Also, the result could be segmented in different sub-sections emphasizing on characteristics of the population, resistance and carriage rate and serotype distribution.

8) It is surprising that the paper is owned by only one author, since an extensive work is performed and more than one centers are involved. I wonder how can we acknowledge others involved in the study.

6. PLOS authors have the option to publish the peer review history of their article (what does this mean?). If published, this will include your full peer review and any attached files.

Reviewer #1: No

Reviewer #2: No

---

## [Author Response · Author response to Decision Letter 0]

16 May 2020

Response to reviewer 1

 Comments Response 

Major issue A major concern is the confusing nature of how the results presented. Many of the finer details can be summarized in the text referring to charts for support, highlight interesting differences for the reader to note. The numerous tables presenting the finer details can be presented in supplementary materials.

 Some representative tables were changed into charts and figures and that their detailed tables were provided as supplementary material as requested

Minor issues 

Page 2, line 35 “DCCs” should be defined on line 27 DCCs was defined in line 27 as first time mentioned in the text

Page 2, line 41 “PCV13 “ should be defined Was changed to “the thirteen valent pneumococcal conjugate vaccine (PCV13)”

Page 2, line 41 PCV13 serotype coverage for Irbid was 65.6%, and for all regions of Madaba 65.3%.” Since the rates are pretty much the same, for the abstract I suggest something similar to “Serotype coverage of the 13 valent pneumococcal vaccine (PCV13) was about 65% for both regions.” Was done or changed as suggested to: Serotype coverage of the thirteen valent pneumococcal conjugate vaccine (PCV13) was about 65% for both regions

Page 2, line 42 “All vaccine serotypes isolated in this study included in the PCV13 showed penicillin resistance from 96.4% to 100% with exception to serotypes 3 and 5 in both cities.” Suggest re-wording to: “Over 96% of isolates with PCV13 serotypes in this study were resistant to penicillin with the exception of serotypes 3 and 5.” Was done or changed in the abstract as suggested to:

Over 96% of isolates with PCV13 serotypes in this study were resistant to penicillin with the exception of serotypes 3 and 5.

Page 3, line 49 “… (S. pneumoniae)…” is not required here

 Done (deleted)

Page 6, line 118 Describe what the level of significance was, ie p<0.05 using 2-tailed values Done as requested to: 

Student t-test was considered for significant differences using 2-tailed values with the level of significance at p<0.05.

Page 7, lines 135-138 If the differences between the two cities are not analysed, the manuscript can be shortened by combining these data for both cities. Keep these details in table 1. Perhaps make table 1 a supplementary table? Table 1 was changed according to the data available in lines 135-138, and Table 1 was changed to Table 1a with detailed number of NP-samples from each center and was put as supplementary table

Page 8 onward page number missing from draft manuscript Page numbers were added 

Line 151 Suggest using a chart to display this information and put Table 2 into supplementary material. A chart as Fig 1. was made to display table 2, and table 2 was put as supplementary material as suggested

Line 165 Table 4 not included in draft. Again, maybe use a chart for this information instead of a table? Table 4 was inserted as Fig 2. As the Table 4 was put as supplementary to the figure

Line 171 If there is no statistical difference, just state as such and combine data for both cities. State detailed data only for the significant differences. The P<0.05 indicates significance? This should be p>0.05? This is true, was changed accordingly

Line 172 I would suggest separating the monthly coverage variations of coverage from that of the antimicrobial resistance data. Keep the same data type together (ie. 1) monthly/seasonal trend of coverage; 2) AMR distributions geographically/temporal). Furthermore, perhaps the monthly trend of AMR is not needed, but present the just annual /regional differences if present? Done as suggested, so that in table 5 was only for the coverage rates on monthly basis for both cities, and in table 6 was for the resistance rates for both cities on monthly basis (Annual rates was shown in other table. Figure 3 and 4 were set for these data and tables 5 and 6 will be as supplementary to show the difference between the two cities

Lines 200-203 Data in Tables 8-10 can be summarized to show the AMR results into a chart instead of the data from tables 6-7? Put detailed counts into supplementary material.

 Tables 8-10 were summarized in one table to avoid repetition of data. Figure 5 was made and the table was set as supplementary as requested 

Line 221-223 Again, for easier reading, if there is no large or interesting differences, just make a general statement with the details. Done as requested

Lines 232-240 Tables 8-10 seem to repeating data covered in previous tables. It is getting confusing as to what differences the read should be noting Tables were restructured, so that no repeated data are available anymore 

Lines 232-240 Are tables 10-11 necessary? Can the data summarized and combined with previous data tables in a more efficient manner? Data for tables 8 and 9 were partly fused into table 10 to avoid repetition of data. Table 10 was then changed into chart and the table was left as supplementary material

Lines 254- 278 Tables 11 – 15 should be summarized into a figure if possible For these tables again to avoid repetition of data, tables 11 and 13 were put in one table, table 12 was deleted, and table 14 and 15 were put in one table. The number of the serotypes and data are huge, so that they could not fit in a chart, hoping you will be ok with this solution

Line 283 Spelling error: “… studding…” ; “studying” ? Spelling error corrected “Studying”

Response to reviewer 2

Major aspects Comments Response 

Point 1 The use of English must greatly improve if the paper is to be published in PLOS ONE. The author needs to take inputs from similar papers in order to improve the overall presentation of his work. The English was improved by a native speaker from England, and more input was inserted from similar papers

Point 2 The “Introduction” section of the manuscript requires extensive revision. First, the author needs to expand the review of literature that is relevant to his study. Second, the aim of the study needs to be properly highlighted and justified. Instead of setting aim in the frame of a simplistic question (reviewer’s personal point of view), I would suggest that the author should attempt to present the key objectives of his study with regards to what is currently known (i.e. literature), thus highlighting the added value of the paper. The introduction was extended with more literature. Regarding the aims of the study were highlighted at the end of the introduction. Since the study group are well known to be in other countries with the highest carriage, and that the DCCs chosen are all in rural areas visited from families mostly with low income and cannot afford the cost of vaccination with the available PCVs. An important issue here is to show the resulted serotypes with more rates belonging to the PCVs harbouring high rates of resistance

Point 3 The inclusion criteria for the children must be clearly stated (for e.g., age group, previous vaccination history considerations and equal population from both the genders). The age group for this study is 2-4 years was chosen by the funding agency, since it is well known worldwide with high carriage rate and according to the world population review reports, this age group count 673.8 thousand or almost 5% of the total population. All children of this study did not have any previous history of PCV vaccination, and this was included in the text. 

Point 4 Details of the demographic surveillance site with independent survey information (Socio-demographic characteristics) should be clearly described. Regarding demographic surveillance and the socio-economic characteristics. Age group is fixed in the study. Children belong to families with low to middle income and all DCCs are in rural areas 

Point 5 There is no mention of the sampling procedures followed or details of the sample size calculation for this study. Regarding sample size. All possible children of age 2-4 years visiting the chosen DCCs in the period of the study were subject to give NP-swabs. Number of NP-swabs was supposed to be 700 but was increased to 1019 at the end of the study to involve most children of this age group in this study

Point 6 To my point of view, materials and method section of the manuscript is written in an amateurish way and does not match the quality standards for being publish in PLOS ONE. The description of the data collection, transportation, laboratory analysis, variable definition and data management & analysis is almost chaotic, while the use of terminology and language is too simplified. Probably a detail flowchart of methodology including the experiment results should be mentioned. The protocol for sampling from 1019 to 341 strain should be elaborated.

 The section Materials and Methods was completely restructured to meet the standards of PLOS ONE as requested

Point 7 To my view, this is not a proper presentation of the statistical results for a survey study. Again, information provided is very hard to follow. The author should consult similar papers to view how the data of a huge sampling study over 2 years should be properly presented and justified. For example, the data given in the manuscript are mainly in tabular form but can be more informative, if represented in the form of graphs or charts. Also, the result could be segmented in different sub-sections emphasizing on characteristics of the population, resistance and carriage rate and serotype distribution. Most of tabular results were changed into charts or figures for better understanding and that repeated results were deleted. Most figures were included in the manuscript and some of the tables were left as supplementary for proper and more informative presentation. Segmentation of the results is also done for each characteristic alone

Point 8 It is surprising that the paper is owned by only one author, since an extensive work is performed and more than one centers are involved. I wonder how can we acknowledge others involved in the study. All participants in this study project have received monthly salaries for sample collection and transfer, and have no impact in writing or data analysis in the paper. Furthermore, Only one paediatric in Jordan have published on pneumococci in addition to me. Another point is that my experience at the National Reference Center for Streptococci in Germany has enabled me to establish a new lab at the German Jordanian University and do most of the work independently.

---

## [Decision Letter · Decision Letter 1]

15 Jun 2020

PONE-D-20-05366R1

Multicenter study of pneumococcal carriage in children 2 to 4 years of age in the winter seasons of 2017-2019 in Irbid and Madaba governorates of Jordan

PLOS ONE

Dear Dr. Al-Lahham,

Thank you for submitting your manuscript to PLOS ONE. After careful consideration, we feel that it has merit but does not fully meet PLOS ONE’s publication criteria as it currently stands. Therefore, we invite you to submit a revised version of the manuscript that addresses the points raised during the review process.

We look forward to receiving your revised manuscript.

Kind regards,

Anirudh K. Singh, Ph.D

Academic Editor

PLOS ONE

Reviewers' comments:

Reviewer's Responses to Questions

**Comments to the Author**

1. If the authors have adequately addressed your comments raised in a previous round of review and you feel that this manuscript is now acceptable for publication, you may indicate that here to bypass the “Comments to the Author” section, enter your conflict of interest statement in the “Confidential to Editor” section, and submit your "Accept" recommendation.

Reviewer #1: (No Response)

Reviewer #2: All comments have been addressed

2. Is the manuscript technically sound, and do the data support the conclusions?

Reviewer #1: Yes

Reviewer #2: Yes

3. Has the statistical analysis been performed appropriately and rigorously? 

Reviewer #1: Yes

Reviewer #2: Yes

4. Have the authors made all data underlying the findings in their manuscript fully available?

Reviewer #1: Yes

Reviewer #2: Yes

5. Is the manuscript presented in an intelligible fashion and written in standard English?

Reviewer #1: No

Reviewer #2: Yes

6. Review Comments to the Author

Reviewer #1: This revised manuscript attempts to improve the presentation of data concerning pneumococcal carriage rates in two cites of Jordan among non-immunized children. Although the paper contains valuable information adding to the knowledge of global dissemination of pneumococcal serotypes and antimicrobial resistance, the author is still having challenges presenting the data in a clear, concise and intelligible manner.

Minor issues:

There are still some grammatical and awkward English translation issues that will need to be addressed by the editor prior to final publication.

Lines 138-144: Some antibiotics are capitalized which should be changed to lower case.

Major Issues:

Line 176 - Figure 1: There seems to be two series of data on the x-axis labelled the same: The first two series “%male carriers 17-18” and “%female carriers 17-18” are labelled the same as series 5 and 6.

Line 185 : Remove table 2.

Figure 3: remove columns for % PCV10 and %PCV7. Keep only %PCV13 and %Carriage.

Figure 4: Should use x-axis as the temporal variable.

Keep Figures 5 and 6, remove figures 1 and 2.

Reviewer #2: This paper experimentally demonstrates that a resistance and carriage rates among the age group 2 to 4 years for a period of two years, reached an alarming rate especially among vaccine types and propose that it can be controlled by pneumococcal conjugate vaccination strategies. The authors have clarified almost all of the questions raised in the previous review. However, the resolutions of the graphs provided could be improved. Overall, this version of manuscript has been sufficiently improved.

7. PLOS authors have the option to publish the peer review history of their article (what does this mean?). If published, this will include your full peer review and any attached files.

Reviewer #1: No

Reviewer #2: No

---

## [Author Response · Author response to Decision Letter 1]

19 Jun 2020

Thank you for the valid feedback received on Monday 15th of June 2020. Below, please find the responses to the reviewers’ comments.

Response to reviewer 1

 Comments Response 

Minor issues: 

Lines 138-144: Some antibiotics are capitalized which should be changed to lower case.

 Capitalized antibiotics were changed into small letters as requested

Major issues: 

Line 176 - Figure 1: There seems to be two series of data on the x-axis labelled the same: The first two series “%male carriers 17-18” and “%female carriers 17-18” are labelled the same as series 5 and 6.

 Figures 1 and 2 were removed as requested in the last point and kept the tables as supplementary material 

Line 185 : Remove table 2. Removed 

Figure 3: Remove columns for % PCV10 and %PCV7. Keep only %PCV13 and % Carriage Data for PCV7 and PCV10 were removed and kept only % carriage and % PCV13 coverage as requested

Figure 4: Should use x-axis as the temporal variable. Done as requested

Keep Figures 5 and 6, remove figures 1 and 2. Keep Figures 5 and 6, remove figures 1 and 2. Figures 1 and 2 were removed as requested and kept Figures 5 and 6. The manuscript text was changed accordingly and kept only the supplementary tables for these figures

Response to reviewer 2 This paper experimentally demonstrates that a resistance and carriage rates among the age group 2 to 4 years for a period of two years, reached an alarming rate especially among vaccine types and propose that it can be controlled by pneumococcal conjugate vaccination strategies. The authors have clarified almost all of the questions raised in the previous review. However, the resolutions of the graphs provided could be improved. Overall, this version of manuscript has been sufficiently improved. Much appreciated for the response. Better resolution of the figures was done as requested

---

## [Editor Report · Decision Letter 2]

23 Jul 2020

Multicenter study of pneumococcal carriage in children 2 to 4 years of age in the winter seasons of 2017-2019 in Irbid and Madaba governorates of Jordan

PONE-D-20-05366R2

Dear Dr. Al-Lahham,

We’re pleased to inform you that your manuscript has been judged scientifically suitable for publication and will be formally accepted for publication once it meets all outstanding technical requirements.

Kind regards,

Anirudh K. Singh, Ph.D

Academic Editor

PLOS ONE
---

## [Editor Report · Acceptance letter]

27 Jul 2020

PONE-D-20-05366R2 

Multicenter study of pneumococcal carriage in children 2 to 4 years of age in the winter seasons of 2017-2019 in Irbid and Madaba governorates of Jordan 

Dear Dr. Al-Lahham:

I'm pleased to inform you that your manuscript has been deemed suitable for publication in PLOS ONE. Congratulations! Your manuscript is now with our production department. 

Kind regards, 

on behalf of

Dr. Anirudh K. Singh 

Academic Editor

PLOS ONE